# Climate Change Mitigation Policies and Co-Impacts on Indigenous Health: A Scoping Review

**DOI:** 10.3390/ijerph17239063

**Published:** 2020-12-04

**Authors:** Rhys Jones, Alexandra Macmillan, Papaarangi Reid

**Affiliations:** 1Te Kupenga Hauora Māori, Faculty of Medical and Health Sciences, University of Auckland, Auckland 1142, New Zealand; p.reid@auckland.ac.nz; 2Department of Preventive and Social Medicine, Division of Health Sciences, University of Otago, Dunedin 9054, New Zealand; alex.macmillan@otago.ac.nz

**Keywords:** climate change, climate policy, environmental justice, Indigenous health, equity

## Abstract

Climate change mitigation policies can either facilitate or hinder progress towards health equity, and can have particular implications for Indigenous health. We sought to summarize current knowledge about the potential impacts (co-benefits and co-harms) of climate mitigation policies and interventions on Indigenous health. Using a Kaupapa Māori theoretical positioning, we adapted a validated search strategy to identify studies for this scoping review. Our review included empirical and modeling studies that examined a range of climate change mitigation measures, with health-related outcomes analyzed by ethnicity or socioeconomic status. Data were extracted from published reports and summarized. We identified 36 studies that examined a diverse set of policy instruments, with the majority located in high-income countries. Most studies employed conventional Western research methodologies, and few examined potential impacts of particular relevance to Indigenous peoples. The existing body of knowledge is limited in the extent to which it can provide definitive evidence about co-benefits and co-harms for Indigenous health, with impacts highly dependent on individual policy characteristics and contextual factors. Improving the quality of evidence will require research partnerships with Indigenous communities and study designs that centralize Indigenous knowledges, values, realities and priorities.

## 1. Introduction

Climate change is a powerful determinant of current and future health for all human populations [1,2]. The nature and magnitude of health impacts and the distribution of those impacts will be determined by the effectiveness of measures to limit the extent of climate change and protect against the adverse health consequences [2]. However, it is clear that the impacts of climate change are inequitable and, without substantial corrective action, will exacerbate existing health inequities between and within countries [3,4]. Disproportionate adverse impacts will be borne by people in low-income countries, and by disadvantaged populations within all countries [5,6].

Indigenous peoples are among those who will be hardest hit by the health impacts of climate change [7,8]. Although there is no universal definition of Indigenous peoples, key features include self-identification, historical continuity with pre-colonial and/or pre-settler societies, and strong links to territories and associated natural resources. Indigenous peoples currently form non-dominant sectors of society with distinct social, economic or political systems, and distinct languages, cultures and beliefs [9,10]. While there is considerable heterogeneity among Indigenous peoples, including in relation to geographical contexts, cultural backgrounds and sociopolitical circumstances, there are common features that confer greater vulnerability to climate change impacts. For many Indigenous populations, these factors are associated with the legacy and ongoing impacts of colonization.

The health and equity impacts of climate change make mitigation efforts critical. In particular, protecting Indigenous health now and into the future requires rapid, effective action to limit global warming to 1.5 degrees Celsius [11,12]. The choice and design of climate policies are also important for Indigenous health and health equity: actions to reduce greenhouse gas (GHG) emissions can have co-benefits and/or co-harms for health, and can either facilitate or hinder progress towards health equity [13,14]. Without explicit attention to Indigenous health and equity, mitigation actions threaten to exacerbate existing inequities [15].

It is therefore imperative that climate mitigation policies and actions are designed to maximize positive impacts and minimize adverse outcomes for Indigenous health. However, evidence on which to base these decisions is limited. Further, the range of possible climate mitigation policy instruments and their multidimensional impacts, including differential impacts on different populations, make this a complex field of study [16]. Co-impacts of climate change mitigation on health can be mediated through a multitude of pathways [17]. Health co-benefits occur through mechanisms such as reduced air pollution from changes in energy generation and transport mode shift, higher levels of physical activity and social contact as a result of increased active transport, reduced intake of foods from animal sources, and healthier indoor environmental conditions due to improved household insulation [18,19]. Adverse health impacts may occur as a result of increased indoor pollution due to reduced household ventilation, increased fuel poverty due to higher energy costs, greater exposure to danger as a result of increased active transport, and affected childhood growth and development from lower animal product consumption [19,20].

Very little has been published on this topic previously. In a recent review of the literature, Markkanen and Anger-Kraavi synthesized evidence about the social co-benefits and co-harms of climate change mitigation policy, including impacts on equity, with a focus on socioeconomic inequities [21]. Their analysis shows that most policies have the potential to produce both co-benefits and co-harms, and can either exacerbate or reduce inequities depending on contextual factors, policy design and policy implementation.

Over and above a more general analysis of climate policy and health equity (related to differential impacts by socioeconomic status and ethnicity), there is also an imperative for an explicit focus on the implications for Indigenous populations. While general understandings of health equity are relevant, the causal pathways leading to inequities for Indigenous peoples have distinctive features as a result of the historical, social and political contexts within which they are situated [22]. These include greater dependence on environmental resources for basic needs, living on marginal land with poorer infrastructure, socioeconomic deprivation, employment inequities, a greater existing burden of disease, poorer access to and quality of health care, and political marginalization [15,23]. Factors such as poverty and racism are clearly important determinants of Indigenous health, but are only intermediate causes. They are fundamentally driven by the ongoing processes and effects of colonization, which in turn determine how these factors confer differential exposure to health risks and benefits [24]. Colonization also constrains the design and diversity of potential climate and health responses, in particular through the systematic suppression of Indigenous knowledge systems and ways of being [25]. Colonization is itself just one manifestation of an exploitative Enlightenment philosophy that has resulted in catastrophic effects on Indigenous peoples and nature [26,27].

Climate change and societal responses to climate change are experienced in qualitatively and quantitatively different ways by Indigenous peoples in comparison with other population groups. Whyte [28] characterizes anthropogenic climate change as an intensification of environmental changes due to colonialism that have disrupted Indigenous peoples’ ways of living for centuries. The specific considerations in relation to the right to health for Indigenous peoples imply a need for an Indigenous-focused approach to this area of research, grounded in Indigenous worldviews and explicitly centering Indigenous lived experiences and realities [29]. Social policy in settler colonial contexts universally fails to adequately consider social determinants of Indigenous health and Indigenous rights [30], which means that generic equity analyses miss important outcomes for Indigenous populations.

This article summarizes current knowledge about the potential impacts (co-benefits and co-harms) of climate mitigation policy and interventions on Indigenous peoples’ health, and factors influencing the direction and magnitude of those impacts, with a view to informing national, regional and local decision making to uphold Indigenous rights including the right to health. In conceptualizing impacts, we use a multidimensional definition of health that draws on Indigenous understandings of well-being. We consider impacts on both health outcomes and the key determinants of health including modifiable social, economic and environmental conditions. Such an approach aligns with understandings of health inequity that include not only differences in health status, but also in important influences on health that are amenable to change, e.g., through social policy [31]. Health equity can be defined as “the absence of systematic disparities in health (or its social determinants) between more and less advantaged social groups” [32] (p. 256) and health inequities as “differences which are unnecessary and avoidable, but in addition are considered unfair and unjust” [33] (p. 431).

## 2. Materials and Methods

### 2.1. Methodology

For this research, we adopted a Kaupapa Māori theoretical positioning. Kaupapa Māori theory arises from Māori (the Indigenous peoples of Aotearoa New Zealand) ontology, epistemology and axiology [34,35], and centralizes Māori worldviews, philosophies and principles [36]. It does not make claims to universal truth or superiority as a research paradigm, but instead seeks to challenge the primacy of ‘conventional’ research methodologies in order to centralize Māori ways of knowing and being [37].

Kaupapa Māori promotes action that is transformative, empowering and liberatory [38]. It seeks to critically examine and expose systems of power that have created and perpetuate social inequities [37]. Kaupapa Māori research methodology does not prescribe or prohibit any particular study designs, methods or tools, but rather can utilize whatever methods are best suited to answer the research question(s). Decisions about methods are informed by Kaupapa Māori principles, including that research should be transformative; promote social justice; be informed by Māori knowledge; focus on critiquing power and privilege; reject cultural-deficit theories; accept diverse Māori realities; and be emancipatory and supportive of decolonization [39].

This research is Indigenous led and two of the authors (R.J. and P.R.) are senior Māori academics, while the third (A.M.) is *Pākehā* (New Zealand settler of European descent) and *tangata Tiriti* (people in NZ by virtue of te Tiriti o Waitangi, the founding treaty between Māori and the British Crown, with a te Tiriti-mandated responsibility for good governance). Collectively, our positionality explicitly acknowledges the central role of colonization in shaping the social, political and economic realities of Indigenous peoples. We assert that it is not possible to understand the implications of climate policy for Indigenous health without situating our review of the evidence within this context. Colonial values and systems also underpin Western academic practice and have powerfully shaped the body of existing research, as well as determining which forms of knowledge are deemed legitimate. While this body of literature has theoretical limitations, there is value in elucidating the issues that are foregrounded within it. We therefore applied a Kaupapa Māori lens in order to review literature with a particular focus on understanding its potential to advance Indigenous health and health equity.

### 2.2. Research Methods

We utilized standard scoping review methods on this Kaupapa Māori theoretical base and report them below according to the relevant sections of the Preferred Reporting Items for Systematic Reviews and Meta-Analyses (PRISMA)-ScR guidelines [40].

#### 2.2.1. Scoping Review Questions

The review sought to answer the following questions:What does the current evidence suggest are important potential impacts (co-benefits and co-harms) of climate change mitigation policies and interventions on Indigenous health?What factors influence the direction and magnitude of these impacts?What are the strengths and limitations of existing research to examine the impacts of climate change mitigation measures on Indigenous health?

#### 2.2.2. Eligibility Criteria

Eligibility of studies was determined according to the criteria detailed below. The review included qualitative, quantitative and mixed methods studies of all study designs. It was limited to English language articles where the full text was available, with no time limit on publication dates. We expected the body of eligible studies specifically relating to Indigenous peoples to be small, and therefore took an inclusive view in how we identified Indigenous populations, as well as including other dimensions of equity. In particular, studies examining the distribution of co-benefits and/or co-harms by ethnicity and socioeconomic status (SES) were included, as they capture dynamics relevant to the etiology of Indigenous/non-Indigenous health inequities. Identifying potentially racist outcomes of policies or interventions (by including studies examining ethnic inequities) is relevant as these racist outcomes are likely to contribute to differential impacts for Indigenous populations. Similarly, in most contexts SES is an important factor mediating Indigenous/non-Indigenous health inequities [41], so actions that have differential impacts by SES are likely to have particular impacts for Indigenous peoples. Studies examining only gender inequities were not included, as there was not considered to be a plausible and consistent mechanism linking gender differences and differences by Indigeneity.

##### ‘Climate Citigation’ Criteria

Inclusion criteria:
Studies of climate mitigation interventions (policies or interventions designed, at least in part, to reduce net greenhouse gas emissions) were eligible.Studies that modeled scenarios associated with climate change mitigation policies or interventions were eligible.Eligible outcome measures included all measures of human health and determinants of health.Quantitative studies were eligible, including both empirical and modeling studies. Under the ‘empirical’ category, randomized controlled trials, non-randomized controlled trials, controlled before–after studies and interrupted time series studies were eligible. In addition, modeling studies that estimated the impact of climate mitigation interventions on health outcomes were eligible.Qualitative studies were eligible.

Exclusion criteria:
Studies that only described climate mitigation policies or interventions, and did not assess impacts on one or more health outcomes or determinants of health, were not eligible.Studies that examined only the health impacts of climate change or of climate-related exposures (e.g., air pollution), and not the health impacts of climate mitigation policy or interventions, were not eligible.Studies with an exclusive focus on climate change adaptation, with no mitigation component, were not eligible.

##### ‘Equity-Focused’ Criteria

Inclusion criteria:
To be eligible, studies must have assessed whether the impacts of climate mitigation measures differ by ethnicity/indigeneity and/or socioeconomic status (SES), or they must have assessed the specific impacts of climate mitigation measures for Indigenous and/or other marginalized populations. The following types of study were eligible:
⚬Studies that reported effect estimates stratified by ethnicity/indigeneity or SES;⚬Studies that reported population-specific effect estimates for Indigenous and/or other marginalized populations;⚬Studies that reported whether or not there was an interaction effect between intervention and the ethnicity/SES variable were eligible.
Indigenous populations were defined according to the approach used by Anderson et al. [42], based on criteria specified by the UNPFII [10].Income, education, employment and housing tenure were eligible measures at individual level. Neighborhood deprivation was an eligible area level measure of SES, and also eligible as a proxy for individual SES.

Exclusion criteria:
Studies that assessed only whether there was confounding by ethnicity/indigeneity/SES (rather than whether intervention effects differed by ethnicity/indigeneity/SES) were not eligible.

#### 2.2.3. Search Strategy

We searched MEDLINE, EMBASE, Web of Science and Scopus from 5 to 8 August 2019. An initial limited search of these databases was conducted, followed by an analysis of text words contained in the title and abstract of identified articles, and of the index terms used. We used these keywords and index terms to develop customized search strategies for each of the databases, based on a validated search strategy for equity-focused reviews [43]. A sample search strategy is provided in Appendix A. In addition, we identified possible evidence sources from the WHO and IPCC websites.

#### 2.2.4. Study Selection

One reviewer (R.J.) screened the titles and abstracts of identified studies to exclude publications that clearly did not meet the inclusion criteria. For each of the remaining articles, the full-text article was retrieved for review and assessed against the inclusion and exclusion criteria. A 10% sample of the full-text articles was assessed by a second reviewer (A.M.).

#### 2.2.5. Data Charting

Data were entered in Microsoft Excel. Given the broad scope of included studies, data charting was an iterative process throughout the review and minor amendments were made to the data variables as required. One reviewer (R.J.) undertook data charting of all included studies, with independent charting of ten randomly selected studies undertaken by a second reviewer (A.M.).

The following data items were collected during the data charting process:Publication characteristics: title, year of publication, study design, country of origin, study setting.Type of study, e.g., modeling, intervention study, qualitative or quantitative methods.Characteristics of mitigation policy or intervention
Context (governmental jurisdiction(s), target population(s));Typology (by IPCC categories) [20]: (i) Policy instrument (e.g., economic instruments, regulatory, government provision of public goods or services); and (ii) Sector (e.g., Energy, Transport, Buildings);Detailed description of the policy measure(s) or intervention(s).Outcomes analyzed
Climate outcomes (e.g., emissions reductions);Social co-benefits and/or co-harms (e.g., air pollution, household costs/savings, employment, food security);Health outcomes (e.g., mortality, morbidity, risk factors).Equity and/or subgroup analyses
Detail of any equity analysis (e.g., by gender, ethnicity, SES);Were Indigenous-specific outcomes reported?Key findings
General findings;Equity findings;Indigenous-specific findings (if applicable).Implications for mitigation policy design and implementation
Factors influencing direction and magnitude of co-benefits/co-harms;Recommendations for promoting pro-equity outcomes.Validity assessment
Internal validity (‘quality’ assessment);External validity (Kaupapa Māori) assessment.

We reported the setting of the policy or intervention according to the relevant policy making or governmental jurisdiction(s), e.g., USA, China, New Zealand. This approach reflects the importance of identifying the national contexts within which climate policy making occurs and is consistent with an emphasis on critiquing systems of power.

#### 2.2.6. Data Synthesis

The data were summarized numerically using descriptive statistical methods, and qualitatively using thematic analysis. The study findings were grouped into different categories of mitigation policy measures, and within each category we summarized the type of settings, populations and study designs, along with the measures used and broad findings. Where we identified a systematic review, we counted the number of studies included in the review that potentially met our inclusion criteria and noted how many studies had been missed by our search.

#### 2.2.7. Validity Assessment

Because of the very heterogeneous and sparse nature of the literature, a detailed assessment of study quality was not undertaken. Internal validity was assessed informally using criteria adapted from CASP checklists [44] for different study types. A greater emphasis was placed on external validity, for which we focused on the potential contribution of studies in relation to Indigenous health and equity, using a set of Kaupapa Māori criteria. This assessment addressed: (i) theoretical/methodological issues (e.g., consistency with Indigenous theoretical positioning and/or critical/decolonial theory); (ii) validity of process (e.g., involvement of Indigenous communities and contribution of Indigenous knowledges and values); and (iii) validity of outcomes (e.g., the extent to which potential impacts of particular relevance to Indigenous peoples were included in the analysis).

## 3. Results

From an initial 3539 citations, we identified 36 published articles, each from separate studies. Thirteen of the initial citations were identified from systematic reviews having been missed by our search; from these citations, one study met the scoping review inclusion criteria and was added. Figure 1 provides details of the study selection process.

Table 1 provides details of the included studies, including governmental jurisdiction, type of study, summary of any Indigenous participation or methodology, policy instrument and sector, a brief description of the policy or intervention, and an indication of positive and/or negative implications for Indigenous health.

Eight studies were carried out in the USA [48,52,53,56,58,70,72,74], seven in the UK [60,68,73,75,77,78,80], three in New Zealand [57,66,67], and two each in China [47,62], India [49,61] and Mexico [50,59]. Other jurisdictions represented were Canada [69], the Dominican Republic [71], Ecuador [64], France [55], Finland [45], Greece [46], Malaysia [65], South Africa [79], Tanzania [63] and Venezuela [51]. Two studies examined policies across multiple jurisdictions [54,76].

The most common study type was modeling (n = 18), which included a range of environmental modeling methods and general equilibrium economic modeling [45,48,52,55,59,60,61,62,65,66,67,68,72,73,77,78,79,80]. Two studies used health impact assessment methods [46,70] and the remainder (n = 16) were empirical studies, including surveys, before/after comparisons, randomized controlled trials and case studies [47,49,50,51,53,54,56,57,58,63,64,69,71,74,75,76].

Table 2 shows how the studies were distributed according to the type of policy instrument and sector.

### 3.1. Indigenous Data

Twelve studies included an identifiable Indigenous population, either as a central focus of the research or as an identifiable subgroup in the analysis [49,50,51,54,56,64,66,67,69,71,74,76]. These studies were carried out in Canada [69], the Dominican Republic [71], Ecuador [64], India [49], Mexico [59], New Zealand [66,67], the USA [56,74], and Venezuela [51], with one study covering Australia and British Columbia, Canada [54] and one examining multiple countries (Brazil, Peru, Cameroon, Tanzania, Indonesia and Vietnam) [76]. Of these studies, nine had an explicit focus on Indigenous communities and/or the policy or intervention was predominantly based in Indigenous communities [50,51,54,56,64,69,71,74,76].

### 3.2. Validity Assessment

The most common type of study involved modeling the impacts of proposed or existing climate policy, with many using general equilibrium macroeconomic modeling. Such models have been subjected to significant and fundamental critiques, including that their foundational assumptions are widely accepted to be wrong (including that the economy tends towards equilibrium, that actors make homogenous rational choices—with an extremely limited understanding of “rational”, and that actors are individualistic and selfish) [81,82], that important differences that determine inequities must be ignored [82] and that large and vital parts of the real economy (e.g., unpaid work and the commons) are absent [81]. These critiques are particularly important to the understanding of impacts on Indigenous health (not just because inequities are missing, but also because of the missing roles of unpaid work, collective ownership and decisions based on collective well-being).

Other studies varied in terms of quality, ranging from well-designed before–after comparison studies to uncontrolled case studies. Almost all employed conventional Western research methodologies and did not incorporate Indigenous theoretical positioning or engage with critical or decolonial theory. Few studies examined potential impacts of particular relevance to Indigenous peoples, such as revitalization of Indigenous languages and knowledge systems, protection of sites of cultural significance or sovereignty.

Only three studies reported being conceived, designed or conducted in partnership with Indigenous communities or informed by Indigenous knowledges and values [56,69,74]. Richards et al. [69] reported on three projects among Inuit peoples in Canada that involved supporting Indigenous communities to conduct their own research. These projects contributed to building community capacity, with significant value placed on Indigenous ways of knowing, doing and being in order to create positive change for their communities. The study by Champion et al. [56] followed key concepts in Navajo philosophy relating to partnership, community consensus, education and critical thinking to develop a culturally-specific framework to evaluate household heating alternatives in the Navajo nation. Sikka et al. [74] examined the outcomes of developing a sustainable biomass energy industry in Alaska Native communities with an explicit emphasis on how the initiative aligned with core cultural values.

### 3.3. Impact by Policy Type

Analysis of economic instruments such as carbon taxes and emissions trading schemes predominantly examined co-impacts mediated through socioeconomic inequities. These instruments were generally shown to be financially regressive, at least in high-income countries and in the absence of mitigating actions [48,54,60,72]. Carbon pricing can lead to improvements in other determinants of health such as air quality [48]. However, these effects can be inequitable, as was observed in California’s cap-and-trade program that resulted in a paradoxical increase in GHG and co-pollutant emissions in disadvantaged neighborhoods [58]. One study examined different methods of revenue recycling as a way to mitigate the regressive impacts of emissions pricing, and found that the most progressive approach was direct household rebates [48]. Revenue can also be recycled in ways which address other aspects of equity, for example programs to reduce energy poverty such as electrification, sustainable housing and rooftop solar for low-income households [79].

One study explicitly considered the impact of carbon pricing policies on Indigenous populations, and identified some important principles for promoting equity [54]. For example, procedural fairness was shown to be a key consideration, including the ability for Indigenous peoples to participate in the process of selecting and designing the policy. Such participation could help to avoid disproportionate impacts from the increased cost of goods and services, and improve access to the measures introduced to mitigate the cost increases. The authors noted the importance of considering the equity implications of the entire policy package, not only in relation to income but also by cultural, ethnic, gender, region and other demographic factors.

Policies based on government provision of public goods or services were most commonly in the buildings sector, with four studies examining housing renovation to improve insulation and/or heating [53,57,73,75]. Three of these studies reported largely positive effects for low income households, including reduced energy use and fuel poverty, increased indoor temperatures and improved health outcomes [53,57,75]. Some adverse impacts on health equity were noted, with one study reporting an increase in general practitioner visits post-intervention [57] and an analysis of the Warm Front program in England identifying a number of challenges including targeting of assistance and increased fuel consumption in some low-income households [75]. The fourth study identified negative impacts on indoor air quality based on modeling of a full retrofit of the UK housing stock, with elevated health risks primarily due to reduced building permeability [73]. This was the only study in this category to explicitly consider equity implications, with generally higher estimated indoor PM_2.5_ concentrations in low-income households in the retrofitted scenario.

Four studies examined regulatory approaches, all of which were situated in the forestry sector and predominantly focused on aspects of REDD (Reducing Emissions from Deforestation and forest Degradation in developing countries) and REDD+ programs (REDD plus sustainable forest management, as well as enhancement of forest carbon stocks) [59,63,64,76]. These tended to have mixed results, with apparently inevitable trade-offs between effectiveness, efficiency and equity [59]. While there were some benefits identified, for example, better governance and potential employment opportunities, two studies demonstrated inequities in relation to knowledge, participation and benefit sharing [63,64]. A further study found no significant contribution of REDD+ to improving well-being or perceived income sufficiency among most communities [76]. Recommendations from these studies examining regulatory approaches included the need to avoid a one-size-fits-all approach, to strengthen participation and ensure greater community involvement throughout the program, and to respect the self-determination of Indigenous communities.

Five studies examined a combination of different types of policy or intervention [45,46,47,56,61] and a further seven were based on scenario modeling rather than a specific policy or policies [52,62,66,68,77,78,80].

The three studies that had documented Indigenous participation and/or incorporation of Indigenous methodologies in the research process contributed distinctive insights [56,69,74]. They examined aspects of interventions and outcome measures that differed in important ways from other studies included in this review. For example, in the study examining climate adaptation and mitigation initiatives in Inuit communities, intergenerational knowledge transmission was identified as a key factor in strengthening community resiliency and promoting the sustainability of these initiatives over time [69]. In relation to home heating in the Navajo nation, as a result of explicitly incorporating community perceptions and cultural values in the analysis, the study arrived at different conclusions than would have been the case if the model had included only the ‘standard’ environmental, health and economic indicators [56]. Among the merits of using wood pellets rather than oil for Alaska Native communities, affirmation of core Indigenous cultural values around preservation of resources for future generations was identified as a key social benefit [74].

### 3.4. Implications for Indigenous Health

Potential co-benefits for Indigenous health identified in the studies arise through a number of different mechanisms, including reduced air pollution; warmer, drier homes; lower energy costs; increased physical activity; improved diets, validation of Indigenous knowledges; and revitalizing and supporting the use of traditional practices. The actual benefits and impacts on Indigenous/non-Indigenous inequities in health depend on a variety of factors relating to policy or intervention design and implementation.

The most important co-harms to Indigenous health identified in this review were due to economically regressive impacts (especially carbon taxes and emissions trading schemes); job losses that disproportionately affect low income and/or Indigenous populations; fuel poverty; and negative impacts on diet (Indigenous dietary systems and traditional food sources were not considered).

## 4. Discussion

This scoping review sought to summarize the current state of knowledge about the co-impacts of climate change mitigation policy and interventions on Indigenous health. In relation to Indigenous peoples and climate change, research attention has tended to focus on adaptation rather than mitigation [23]. Nonetheless, there are examples in the published literature of local-level mitigation initiatives that have been evaluated with regard to outcomes for Indigenous populations [51,63,71]. However, to our knowledge this is the first systematic review of climate change mitigation policy with respect to Indigenous health.

This scoping review was inclusive of a broad range of policy instruments, interventions and study designs. We adapted a search strategy that had been developed and validated for reviews investigating whether the effects of interventions differ by ethnicity or socioeconomic status. In keeping with a Kaupapa Māori methodological approach, we explicitly analyzed how well studies addressed issues of importance to Indigenous peoples. However, given the wide scope of the review and the heterogeneity of terminology used in article titles, abstracts and keywords, our search is likely to have missed relevant studies. The review also had an emphasis on publications in the peer-reviewed literature, which may have led to the exclusion of potentially eligible studies, for example those disseminated as unpublished evaluations and reports. The search may also have missed studies reported in non-indexed journals. These factors may explain the dearth of Indigenous-centered research identified by our search strategy. However, there is utility in examining the published literature as this is likely to be an important source of information for policy makers.

The heterogeneity of evidence sources, including the range of study designs, contexts, policy and intervention types, and approaches to equity analysis make it difficult to draw specific conclusions about the relative merits of particular interventions. Climate mitigation draws on numerous different policy instruments across many different sectors [20], with myriad possible co-impacts on health and on the determinants of health [17]. Indeed, our analysis indicates a range of possible impacts on Indigenous health, both positive and negative. There does not appear to be a clear pattern based on policy or intervention type, with specific features and contextual factors likely to be important in determining impacts. Rather than specifically guiding policy selection or design, the breadth of sources considered in this study allow the identification of general principles that can inform climate mitigation efforts in order to contribute to Indigenous health equity.

With its particular emphasis, our review complements existing reviews and adds novel insights. For example, the recent synthesis of evidence by Markkanen and Anger-Kraavi [21] included only one of the studies in our sample [63]. Like our review, it showed that most climate change mitigation policies can produce both co-benefits and co-harms, and can be either pro- or anti-equity depending on a range of factors. It also found that the adverse equity impacts of climate policies can be mitigated, but noted that this requires a deliberate, carefully planned approach with community engagement. Our review had a greater focus on critically examining the literature with respect to its implications for Indigenous health.

Most studies in our review did not explicitly consider Indigenous health equity. Existing evidence therefore comes from research that is generally not designed to centralize Indigenous considerations. Drawing conclusions about potential impacts on Indigenous health requires a degree of inference based on factors such as differences by socioeconomic status, which leave out a range of specific considerations for Indigenous peoples. The outcome of this information gap will be climate mitigation actions that fail to fully account for the unique realities of Indigenous peoples or address the specific determinants of inequity, especially those resulting from ongoing colonization. Policies and interventions are therefore likely to drive further inequity, which is a breach of Indigenous rights [83].

Further, the vast majority of the evidence identified in this review relates to climate mitigation that operates within existing social, political and economic systems. This is problematic from a Kaupapa Māori perspective given the integral role these systems play in perpetuating not only climate change [3] but processes of colonization, marginalization and exploitation that drive Indigenous health inequities. Genuine climate solutions must seek to disinvest from institutions and systems that are complicit in fueling the climate crisis, and instead must be grounded in different ways of knowing, doing and being that reflect Indigenous values [84].

These gaps in existing research and policy mean that there is insufficient evidence to inform climate mitigation action that can uphold Indigenous rights and contribute to health equity. In order to address these gaps, research must centralize Indigenous worldviews, methodologies, realities and priorities. As underlined by our analysis of the three studies in this review that adopted Indigenous methodologies, such research examines the issues through a distinctive lens that generates unique and important insights. These studies highlighted issues not considered in other studies, including aspects of interventions such as intergenerational knowledge transmission and incorporation of cultural values, as well as critical contextual factors related to ongoing experiences of colonization. Brugnach et al. [85] provide a conceptual framework to support the participation of Indigenous communities in climate mitigation policy and decision making, emphasizing the recognition of Indigenous knowledges and the need for power-sharing. As the impacts of climate mitigation measures on Indigenous health equity are highly dependent on the specific characteristics of the intervention and relevant contextual factors, these generic principles of engagement are critical in ensuring that Indigenous realities and priorities are addressed and that equity issues are centered.

## 5. Conclusions

There is a dearth of information about the co-benefits and co-harms of climate mitigation policy for Indigenous health. Much of the evidence that currently exists is from generic equity analyses and limited Western perspectives on relevant outcomes, which has serious limitations in relation to informing future climate policy. Very few studies have used Indigenous or decolonial research methodologies, which influences the philosophical basis of the research, the questions asked, methods used, type of outcomes examined, interpretation of findings and translation of knowledge into action.

Despite the complexity of this field of research, it is possible to improve the quality of evidence about co-impacts on Indigenous health in order to inform pro-equity climate mitigation. This will require partnership with Indigenous communities, recognition and privileging of Indigenous knowledges and study design that fully embeds Indigenous values, realities and priorities. Fundamentally, sharing of power, both in the research process and in the conception, design and implementation of climate policy and interventions, will be essential for Indigenous rights and health equity.

## Figures and Tables

**Figure 1 ijerph-17-09063-f001:**
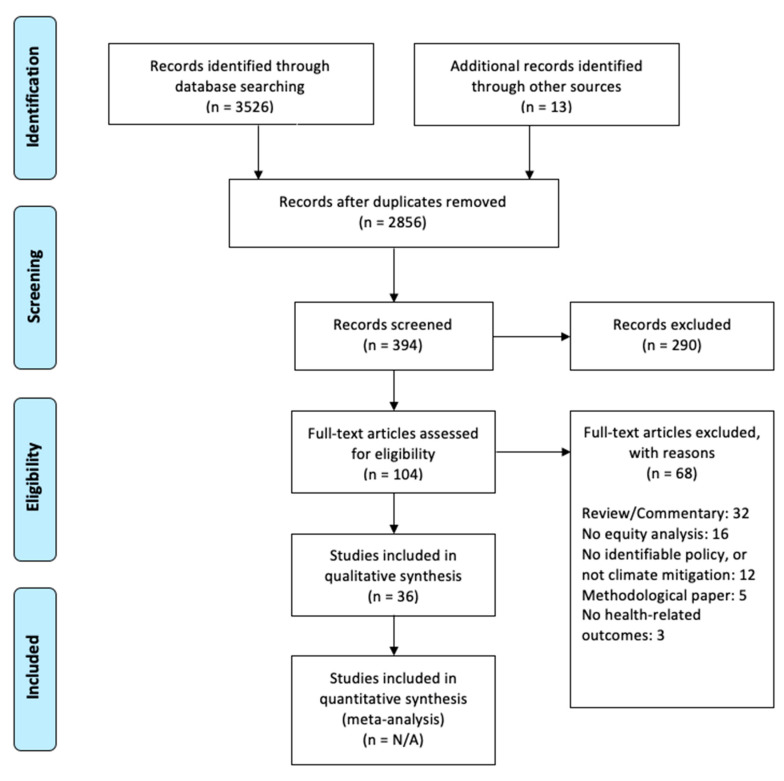
Preferred Reporting Items for Systematic Reviews and Meta-Analyses (PRISMA) flow diagram for study inclusion.

**Table 1 ijerph-17-09063-t001:** Details of included studies.

Citation	Jurisdiction	Study Type	Indigenous Population/Methodology ^1^	Policy Instrument	Sector	Description of Policy or Intervention	Implications for Indigenous Health ^2^
Asikainen 2017 [45]	Finland	Modeling	No	Multiple	Multiple	Energy efficiency and renewable energy	–
Bailey 2019 [46]	Greece	Health impact assessment	No	Multiple	Buildings	Reducing residential wood burning	↑
Barrington-Leig 2019 [47]	China	Cross-sectional survey	No	Multiple	Buildings	Reducing household coal use	↑ ↓
Barron 2018 [48]	USA	Modeling	No	Economic Instruments —Taxes	Multiple	Carbon tax	↑ ↓
Basu 2014 [49]	India	Cross-sectional survey	Population	Government Provision of Public Goods or Services	AFOLU ^3^	Agro-forestry	↑
Berrueta 2017 [50]	Mexico	Case study	Population	Government Provision of Public Goods or Services	Buildings	Improved cookstoves	↑
Bilbao 2010 [51]	Venezuela	Experiment	Population	Government Provision of Public Goods or Services	AFOLU	Indigenous use of fire for forest protection	↑
Boyce 2013 [52]	USA	Modeling	No	Scenario rather than specific policy/ies	Industry	Reducing industrial emissions	↑
Breysse 2011 [53]	USA	Before–after comparison	No	Government Provision of Public Goods or Services	Buildings	Renovation of low-income housing	↑
Bubna-Litic 2012 [54]	Canada and Australia	Case studies	Population	Economic Instruments —Taxes and Tradable Allowances	Multiple	Compares carbon pricing policies	↑ ↓
Caillavet 2016 [55]	France	Modeling	No	Economic Instruments —Taxes	Multiple	Food taxes	↓
Champion 2017 [56]	USA	Mixed methods	Population Methodology	Multiple	Buildings	Home heating in Navajo nation	↑
Chapman 2009 [57]	New Zealand	Cost-benefit analysis of cluster randomized trial	No	Government Provision of Public Goods or Services	Buildings	Home insulation in low-income areas	↑
Cushing 2018 [58]	USA	Before–after comparison	No	Economic Instruments —Tradable Allowances	Industry	Cap-and-trade program	↓
Dyer 2012 [59]	Mexico	Modeling	No	Regulatory Approaches	AFOLU	REDD+ ^4^	↑ ↓
Feng 2010 [60]	United Kingdom	Modeling	No	Economic Instruments —Taxes	Multiple	GHG emissions taxes	↓
Garg 2011 [61]	India	Modeling	No	Multiple	Multiple	Reducing air pollution	↑
Ji 2015 [62]	China	Modeling	No	Scenario rather than specific policy/ies	Transport	Increased EV use	↓
Khatun 2015 [63]	Tanzania	Case study	No	Regulatory Approaches	AFOLU	PFM ^5^ REDD+	↑
Krause 2013 [64]	Ecuador	Cross-sectional survey	Population	Regulatory Approaches	AFOLU	REDD+	↑ ↓
Li 2017 [65]	Malaysia	Modeling	No	Economic Instruments —Subsidies	Multiple	Removing fossil fuel subsidies	↑ ↓
Lindsay 2011 [66]	New Zealand	Modeling	Population	Scenario rather than specific policy/ies	Transport	Transport mode shift	↑
Ni Mhurchu 2015 [67]	New Zealand	Modeling	Population	Economic Instruments —Taxes	Multiple	Food tax including GHG	↑
Reynolds 2019 [68]	United Kingdom	Modeling	No	Scenario rather than specific policy/ies	Multiple	Dietary changes	↑ ↓
Richards 2019 [69]	Canada	Case studies	Population Methodology	Voluntary Actions	Multiple	Community initiatives	↑
Richardson 2012 [70]	USA	Health impact assessment	No	Economic Instruments —Tradable Allowances	Multiple	Cap-and-trade program	↑ ↓
Sánchez 2017 [71]	The Dominican Republic	Case studies	Population	Government Provision of Public Goods or Services	Energy	Micro- hydropower systems	↑
Shammin 2009 [72]	USA	Modeling	No	Economic Instruments —Tradable Allowances	Multiple	Cap-and-trade program	↑ ↓
Shrubsole 2016 [73]	United Kingdom	Modeling	No	Government Provision of Public Goods or Services	Buildings	Energy efficiency retrofitting of homes	↓
Sikka 2013 [74]	USA	Case study	Population Methodology	Voluntary Actions	Energy	Transition to biomass energy	↑ ↓
Sovacool 2015 [75]	England	Case study	No	Government Provision of Public Goods or Services	Buildings	Energy efficiency retrofitting of homes	↑ ↓
Sunderlin 2017 [76]	Multiple jurisdictions	Longitudinal (before–after) survey	Population	Regulatory Approaches	AFOLU	REDD+	–
Tainio 2017 [77]	England	Modeling	No	Scenario rather than specific policy/ies	Multiple	Diet and physical activity scenarios	↑ ↓
Williams 2018 [78]	Great Britain	Modeling	No	Scenario rather than specific policy/ies	Multiple	Modeling of energy scenarios	↑ ↓
Winkler 2017 [79]	South Africa	Modeling	No	Government Provision of Public Goods or Services	Multiple	Options for recycling carbon tax revenue	↑
Woodcock 2018 [80]	England	Modeling	No	Scenario rather than specific policy/ies	Transport	Cycling mode share scenarios	↑

^1^ Indicates studies that included an identifiable Indigenous population (‘Population’) and/or used Indigenous methodologies (‘Methodology’). ^2^ Potential implications of the policy or scenario for Indigenous health, either neutral (–), positive (↑), negative (↓) or mixed (↑ ↓). ^3^ AFOLU = Agriculture, Forestry and Other Land Use. ^4^ REDD+ = Reducing Emissions from Deforestation and forest Degradation in developing countries, plus sustainable forest management and enhancement of forest carbon stocks. ^5^ PFM = Participatory Forest Management.

**Table 2 ijerph-17-09063-t002:** Distribution of studies according to policy instrument and sector.

	AFOLU ^1^	Buildings	Energy	Industry	Transport	Multiple Sectors	Totals
Economic instruments				1		8	9
Government provision	2	5	1			2	10
Regulatory approaches	4						4
Voluntary actions			1				1
Multiple policy types		3				2	5
Scenario				1	3	3	7
Totals	6	8	2	2	3	15	36

^1^ AFOLU = Agriculture, Forestry and Other Land Use.

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
