# Peer review of "Climate Change Mitigation Policies and Co-Impacts on Indigenous Health: A Scoping Review"

_ijerph, 2020, doi:10.3390/ijerph17239063_

Round 1

Reviewer 1 Report

This scoping review summarises peer-reviewed published literature that examined the benefits and harms of climate mitigation policies in relation to Indigenous health. This review utilised standard scoping review protocol, following PRISMA-ScR guidelines. It also used Kaupapa Māori as a basis for theoretical grounding. The review concludes that most outcomes do not consider Indigenous health equity and that the main limitation of current literature available in this field is that potentially relevant outcomes are dominated by Western perspectives. The manuscript concludes that more partnerships between researchers working within Indigenous-related fields and Indigenous communities should be formed to improve the effect of climate mitigation policy on Indigenous health. This scoping review is important for identifying existing literature about the impact of climate mitigation policies on Indigenous populations and as stated by the authors, it represents a novel contribution to the literature.

Summary of Strengths

Overall, this manuscript is well-written, succinct, and easily readable. I appreciated that the authors included their positionality. The level of detail in the description of the methodology is excellent. More specifically, the authors identify clear inclusion and exclusion criteria and present their findings in a series of tables that are consistently formatted and communicate findings clearly and effectively. The content of this manuscript is also very relevant to the International Journal of Environmental Research and Public Health. Furthermore, any limitations of this study were disclosed and were addressed within the discussion section.

Summary of Weaknesses

In addition to the specific feedback below, there are some revisions needed to this manuscript before it could be recommended for publication. While the authors’ purpose clearly guides the results and discussion in the manuscript, the authors do not clearly answer their scoping review questions. For example, it is not clear by the end of the manuscript which impacts of climate change mitigation policies are most important for Indigenous health. This applies to all four questions that are posed. While the lack of literature in the subject area certainly limits the authors’ ability to answer these questions, this manuscript would benefit from one, overarching question that is more directly connected with the purpose stated in the introduction (i.e. to summarise the co-benefits and co-harms of these policies in regard to Indigenous health). Furthermore, key terms, such as “Indigenous”, are not defined, making it difficult to ascertain who is considered part of an Indigenous population for the purposes of the review (for example, some scholars argue that many of the 56 ethnic groups in nation-states like China are Indigenous, while others argue that no one in nation-states like China are Indigenous). Furthermore, even though the authors indicate that they are using an Indigenous theoretical positioning and are reviewing findings for consistency with Indigenous and decolonial theories, terms such as “Canada” and “United States” are used even though these terms represent oppressive, colonial nation-states that undermine Indigenous nation-states. Thus, similar to Aotearoa, the term Turtle Island should be used to also describe the land now claimed by the Canadian and American governments.  Other terms, like “health” and “wellbeing” as well as “equity” and “equality” appear to be used interchangeably and require clarification. There are also minor inconsistencies with spelling (e.g. both “modeling” and “modelling” are used in the manuscript; data is sometimes singular, sometimes plural) that require minor revisions.

Additional Comments:

Abstract:

Page 1, Line 13-14: You use the term “wellbeing”, but then shift to using the term “health” for the remainder of the abstract. Thus, consider changing the word “wellbeing” to “health”, or differentiating the terms for the reader.

Page 1, Line 14-16: In your introduction, you state that you will “examine potential impacts (co-benefits and co-harms)…”whereas you state you will “summarize…potential impacts (co-benefits and co-harms)” in your abstract. Revise for consistency.

Page 1, Line 20-21: The WTO and the OECD both acknowledge that there is no definition for what constitutes a “developed” nation-state. How did you define “developed” nation-states? If it is based on income, consider using an income classification system (low, middle, high-income) for clarity.

Introduction:

Page 1, Line 37: What is a “poorer” country?

Page 1, Line 44 to Page 2, Line 46: Provide a citation for this statistic.

Page 2, Line 49-52: In these sentences, you use the terms “well-being”, “equity”, and “health” in different groupings (e.g. “health and wellbeing”, “health equity”, “wellbeing and equity”, and “health”.) Revise any terms that are used interchangeably and define the terms that are used in the manuscript.

Page 2, Line 60: Change “adverse side-effects” to “co-harms” for clarity.

Page 2, Line 62-73: This paragraph is repetitive with the paragraph on Page 1, Lines 38-43.

Page 2, Line 91: Inequality or inequity? Based on what has been described in your introduction, it appears that you are referring to inequity. In some cases, these terms appear to be used interchangeably (e.g. On Page 3, you use “inequity” on Line 138 then change to “inequality” on Lines 139 and 142). This should be revised throughout the manuscript.

Page 2, Line 84: Add “interventions” as they are examined in addition to policies for climate mitigation based on your inclusion criteria. This should also be added, where appropriate, throughout the manuscript.

Materials and Methods:

General: Although you acknowledge the limitation of search terms, key terms like American Indian, Inuit, Métis are missing that would impact the types of articles retrieved in the review.

Page 3, Line 123-130: Some of these questions do not appear as part of your purpose outlined in the Introduction, whereas part of your purpose in the Introduction (“with a view to informing national, regional and local decision making to uphold Indigenous rights including the right to health.”, is not reflected in these questions. The questions that guide your scoping review should be more closely linked with your stated purpose.

Page 3, Line 137: By capturing studies examining co-benefits and co-harms according to SES, what overlap exists with the review by Markkanen and Anger-Kraavi? Your findings should be differentiated in the Discussion.

Page 4, Line 143-145: This sentence is unclear and should be revised.

Page 4, Line 172-176: How did you define Indigenous peoples, or indigeneity, for the purposes of the review?

Page 4, Line 190: Given the questions posed in your review, why were peer-reviewed, non-indexed Indigenous journals not included as part of your search strategy?

Page 5, Line 239-241: You state that when you identified a systematic review, you noted how many studies had been missed by your search (and which ones had been included). Were there any studies that met inclusion criteria and were missed by your search? Were any studies added after reviewing the included articles in these systematic reviews?  This requires clarification, even if no additional studies from these systematic reviews were added.

Results:

Page 7, Table 1: Like New Zealand, the names “Canada”, “United States” represent colonial nation-states and colonial governments that continue to undermine Indigenous sovereignty in those regions. Both Canada and the United States are a part of Turtle Island, a vast landmass that has been artificially separated by colonial powers now occupying those territories. Perhaps these locations (and others that this reviewer is unaware of) should be recognised by names other than those given by oppressive colonial forces similar to Aotearoa, consistent with decolonial theory and statements made in your introduction that “colonial values and systems also underpin Western academic practice and have powerfully shaped the body of existing research, as well as determining which forms of knowledge are deemed legitimate.”

Page 10, Line 320 – Page 11, Line 379: Many sentences in this section are discussion points that should not be included in the results sections. For example, Page 11, Line 337-339. These sentences should be removed for clarity.

Page 11, Line 359: Add a comma (,) after “…for example”.

Discussion:

General: Given that many of the included studies in your scoping review organised results by SES and given that you mention the review by Markkanen and Anger-Kraavi in the introduction, any overlap or differences in findings with this review should be mentioned in your discussion.

Page 12, Line 395-396: Add citations of the examples from the published literature.

Page 12, Line 416-426: This paragraph repeats the same details provided in the Results, specifically subsections “Validity Assessment” and “Impact by Policy Type”. This paragraph should be removed, with an expanded discussion on the potential “implications for Indigenous health”, as outlined in Table 1.

Page 13, Line 427-435: Although this paragraph provides some insights about how the available literature would effect decision making, given that your purpose statement in the introduction ends with: “…with a view to informing national, regional and local decision making to uphold Indigenous rights including the right to health”, your discussion would benefit from explicitly addressing how the evidence would (or even if it could) inform decision-making at any level of governance.

Author Response

Response to Reviewer 1 comments

We would like to thank you very much for the thorough and helpful feedback on our manuscript. In the table below we have provided details of the revisions in the manuscript and explained our responses to the reviewer comments.

Reviewer comment

Authors’ response

While the authors’ purpose clearly guides the results and discussion in the manuscript, the authors do not clearly answer their scoping review questions. For example, it is not clear by the end of the manuscript which impacts of climate change mitigation policies are most important for Indigenous health. This applies to all four questions that are posed. While the lack of literature in the subject area certainly limits the authors’ ability to answer these questions, this manuscript would benefit from one, overarching question that is more directly connected with the purpose stated in the introduction (i.e. to summarise the co-benefits and co-harms of these policies in regard to Indigenous health).

key terms, such as “Indigenous”, are not defined, making it difficult to ascertain who is considered part of an Indigenous population for the purposes of the review (for example, some scholars argue that many of the 56 ethnic groups in nation-states like China are Indigenous, while others argue that no one in nation-states like China are Indigenous).

We have added an explanation to the introduction about defining Indigenous peoples (lines 39-43).

even though the authors indicate that they are using an Indigenous theoretical positioning and are reviewing findings for consistency with Indigenous and decolonial theories, terms such as “Canada” and “United States” are used even though these terms represent oppressive, colonial nation-states that undermine Indigenous nation-states. Thus, similar to Aotearoa, the term Turtle Island should be used to also describe the land now claimed by the Canadian and American governments.

See detailed response below.

Other terms, like “health” and “wellbeing” as well as “equity” and “equality” appear to be used interchangeably and require clarification.

See detailed responses below.

There are also minor inconsistencies with spelling (e.g. both “modeling” and “modelling” are used in the manuscript; data is sometimes singular, sometimes plural) that require minor revisions.

These have been corrected.

Page 1, Line 13-14: You use the term “wellbeing”, but then shift to using the term “health” for the remainder of the abstract. Thus, consider changing the word “wellbeing” to “health”, or differentiating the terms for the reader.

Change made (line 14).

Page 1, Line 14-16: In your introduction, you state that you will “examine potential impacts (co-benefits and co-harms)…”whereas you state you will “summarize…potential impacts (co-benefits and co-harms)” in your abstract. Revise for consistency.

We have changed the wording in the introduction (line 119).

Page 1, Line 20-21: The WTO and the OECD both acknowledge that there is no definition for what constitutes a “developed” nation-state. How did you define “developed” nation-states? If it is based on income, consider using an income classification system (low, middle, high-income) for clarity.

Amended to low-, middle- and high-income throughout the manuscript.

Page 1, Line 37: What is a “poorer” country?

This has been changed to “low-income countries” (lines 36-37).

Page 1, Line 44 to Page 2, Line 46: Provide a citation for this statistic.

Citations have been added to support this statement (line 54).

Page 2, Line 49-52: In these sentences, you use the terms “well-being”, “equity”, and “health” in different groupings (e.g. “health and wellbeing”, “health equity”, “wellbeing and equity”, and “health”.) Revise any terms that are used interchangeably and define the terms that are used in the manuscript.

We have used the words “equity”, “inequity” and “inequities” throughout the manuscript.

We have used the word “health” throughout the manuscript, with the word “wellbeing” used only when referring to other sources.

Page 2, Line 60: Change “adverse side-effects” to “co-harms” for clarity.

Changed (line 76).

Page 2, Line 62-73: This paragraph is repetitive with the paragraph on Page 1, Lines 38-43.

We have moved part of the last sentence of the earlier paragraph (“... greater dependence on environmental resources for basic needs, living on marginal land with poorer infrastructure, socioeconomic deprivation, employment inequities, a greater existing burden of disease, poorer access to and quality of health care, and political marginalization.”) to the later paragraph. That leaves the earlier paragraph to simply state the problem (that Indigenous peoples are disproportionately affected), while the later paragraph examines the underlying mechanisms and explains why a specific focus on Indigenous peoples is justified. (Lines 83-86)

Page 2, Line 91: Inequality or inequity? Based on what has been described in your introduction, it appears that you are referring to inequity. In some cases, these terms appear to be used interchangeably (e.g. On Page 3, you use “inequity” on Line 138 then change to “inequality” on Lines 139 and 142). This should be revised throughout the manuscript.

We have changed “disparity or inequality” to “inequity” (line 126).

As noted above, we have used the words “equity”, “inequity” and “inequities” (rather than “equality”, “inequality” and “inequalities”) throughout the manuscript.

Page 2, Line 84: Add “interventions” as they are examined in addition to policies for climate mitigation based on your inclusion criteria. This should also be added, where appropriate, throughout the manuscript.

We have added “and interventions” to this sentence (line 120), and have addressed this throughout the manuscript by either adding “interventions” or using a generic word to cover both policy and interventions (e.g. efforts, actions, measures).

Materials and Methods, General: Although you acknowledge the limitation of search terms, key terms like American Indian, Inuit, Métis are missing that would impact the types of articles retrieved in the review.

This is a limitation and would have impacted the coverage of articles.

Page 3, Line 123-130: Some of these questions do not appear as part of your purpose outlined in the Introduction, whereas part of your purpose in the Introduction (“with a view to informing national, regional and local decision making to uphold Indigenous rights including the right to health.”, is not reflected in these questions. The questions that guide your scoping review should be more closely linked with your stated purpose.

We have clarified these questions to reflect the key foci of the scoping review (lines 169-174).  They now align with the purpose outlined in the introduction.

The quoted text (“with a view to informing national, regional and local decision making to uphold Indigenous rights including the right to health.”) is more an overarching purpose of the research rather than a specific question to be answered.

Page 3, Line 137: By capturing studies examining co-benefits and co-harms according to SES, what overlap exists with the review by Markkanen and Anger-Kraavi? Your findings should be differentiated in the Discussion.

We have added a paragraph in the Discussion differentiating these findings (lines 521-528).

Page 4, Line 143-145: This sentence is unclear and should be revised.

We have revised this sentence (lines 188-190): “Studies examining only gender inequities were not included, as there was not considered to be a plausible and consistent mechanism linking gender differences and differences by Indigeneity.”

Page 4, Line 172-176: How did you define Indigenous peoples, or indigeneity, for the purposes of the review?

This has been clarified in the Methods. We have added details about how Indigenous populations were defined for the purposes of the scoping review (lines 180 and 243-244).

Page 4, Line 190: Given the questions posed in your review, why were peer-reviewed, non-indexed Indigenous journals not included as part of your search strategy?

Thank you for pointing this out – we have acknowledged this as a limitation in the Discussion (lines 507-509).

Page 5, Line 239-241: You state that when you identified a systematic review, you noted how many studies had been missed by your search (and which ones had been included). Were there any studies that met inclusion criteria and were missed by your search? Were any studies added after reviewing the included articles in these systematic reviews?  This requires clarification, even if no additional studies from these systematic reviews were added.

13 additional studies from systematic reviews were included in the abstract review, of which one met inclusion criteria and was added. We have added a sentence in the results section to explain this (lines 326-327). We hope this is appropriately represented in the flow chart on page 7.

Page 7, Table 1: Like New Zealand, the names “Canada”, “United States” represent colonial nation-states and colonial governments that continue to undermine Indigenous sovereignty in those regions. Both Canada and the United States are a part of Turtle Island, a vast landmass that has been artificially separated by colonial powers now occupying those territories. Perhaps these locations (and others that this reviewer is unaware of) should be recognised by names other than those given by oppressive colonial forces similar to Aotearoa, consistent with decolonial theory and statements made in your introduction that “colonial values and systems also underpin Western academic practice and have powerfully shaped the body of existing research, as well as determining which forms of knowledge are deemed legitimate.”

Given the focus of the scoping review we believe the key variable to report is the governmental jurisdiction, rather than the Indigenous territory (for those places where there is an Indigenous population). We have changed this column in Table 1 accordingly, and added an explanatory note in the Methods (lines 303-306).

Page 10, Line 320 – Page 11, Line 379: Many sentences in this section are discussion points that should not be included in the results sections. For example, Page 11, Line 337-339. These sentences should be removed for clarity.

These sentences have either been removed or reworded to make it clear that they are reporting results (lines 406-472).

Page 11, Line 359: Add a comma (,) after “…for example”.

Done (line 451).

Discussion, General: Given that many of the included studies in your scoping review organised results by SES and given that you mention the review by Markkanen and Anger-Kraavi in the introduction, any overlap or differences in findings with this review should be mentioned in your discussion.

We have added a paragraph to this effect in the Discussion (lines 521-528).

Page 12, Line 395-396: Add citations of the examples from the published literature.

Citations have been added (line 496).

Page 12, Line 416-426: This paragraph repeats the same details provided in the Results, specifically subsections “Validity Assessment” and “Impact by Policy Type”. This paragraph should be removed, with an expanded discussion on the potential “implications for Indigenous health”, as outlined in Table 1.

This paragraph has been removed.

We have also expanded the discussion on the potential implications for Indigenous health in the Discussion (lines 515-518).

Page 13, Line 427-435: Although this paragraph provides some insights about how the available literature would effect decision making, given that your purpose statement in the introduction ends with: “…with a view to informing national, regional and local decision making to uphold Indigenous rights including the right to health”, your discussion would benefit from explicitly addressing how the evidence would (or even if it could) inform decision-making at any level of governance.

We have revised the last paragraph in the discussion to explain that the available evidence is limited in the extent to which it can inform decision-making (lines 571-584).

Reviewer 2 Report

The submitted manuscript draws attention to an urgent issue of co-impacts of climate change mitigation policy on health of the Indigenous peoples, specifically of the current state of the related knowledge and the way how it is researched. This approach re-opens the question of the shifting paradigm towards participatory research and research power of the indigenous knowledge and knowledge systems.

It points on the limits of the ontology, epistemology and axiology of the Western research paradigm and academic practice perceived as a universal truth.

This manuscript present a well-designed research and demonstrates very important directions of the future research conceding Indigeneity approach. It is also informative for the reflection of the climate change mitigation policy.

In the introduction chapter, health impacts of climate change mitigation (generally) should be briefly described / classified. The same pays for the situation regarding the indigenous peoples’ health, their traditional knowledge system and lifestyle reflected in the use of medicinal plants and indigenous healing techniques should be mentioned as well as dependence of their health on the local natural resources. The heterogenous situation of the indigenous peoples’ health should be discussed, mainly which regards geopolitical factors and differences between urban and rural areas. 

At the end of the results chapter, design of some comprehensive scheme illustrating the key research findings would be helpful. 

I strongly recommend to publish this manuscript!

Author Response

Response to Reviewer 2 comments

We would like to thank you very much for the helpful feedback on our manuscript. In the table below we have provided details of the revisions in the manuscript and explained our responses to the reviewer comments.

Reviewer comment

Authors’ response

In the introduction chapter, health impacts of climate change mitigation (generally) should be briefly described / classified.

We have added a description of the health co-impacts of climate change mitigation (lines 63-71).

The same pays for the situation regarding the indigenous peoples’ health, their traditional knowledge system and lifestyle reflected in the use of medicinal plants and indigenous healing techniques should be mentioned as well as dependence of their health on the local natural resources. The heterogenous situation of the indigenous peoples’ health should be discussed, mainly which regards geopolitical factors and differences between urban and rural areas.

We have added a point about Indigenous peoples’ greater dependence on environmental resources for basic needs (line 83) and have referred to the heterogeneity of Indigenous peoples (lines 43-44).

At the end of the results chapter, design of some comprehensive scheme illustrating the key research findings would be helpful.

We are interested in this idea, but are not clear exactly what the reviewer is asking for. We would be grateful for clarification if this is deemed to be an important issue to address.

Reviewer 3 Report

This is an excellent and very well-written manuscript that scopes the evidence about the inter-related benefits and arms of climate change mitigation policy for Indigenous populations and their health and well-being. The authors have meticulously synthesised evidence from 36 studies and identified important gaps to be addressed through policy and future research.  

Introduction: Strong, comprehensive, flows well.

Methods: Strong justification of theoretical framework used. In the search strategy. Please state whether the databases were searched from database inception.

Results: Quite often the authors say “Four studies” or “Three studies” but these studies are not cited at the end of the sentence.

Discussion: Excellent

Conclusion: Appropriate

Tables and Figures: Appropriate

Author Response

Response to Reviewer 3 comments

We would like to thank you very much for the helpful feedback on our manuscript. In the table below we have provided details of the revisions in the manuscript and explained our responses to the reviewer comments.

Reviewer comment

Authors’ response

In the search strategy. Please state whether the databases were searched from database inception.

Databases were searched from database inception. We have stated that there was no time limit on publication dates (line 178).

Results: Quite often the authors say “Four studies” or “Three studies” but these studies are not cited at the end of the sentence.

These studies have now been cited.

Reviewer 4 Report

This is a well written paper on a topic that is under-researched but nevertheless absolutely critical to environmental health and climate mitigation, that of  the co-benefits and co-harms indigenous health and social systems. From an initial data base of 3539 citations 36 studies were extracted for review that conformed to the 'eligibility criteria' which focused on climate mitigation and indigenous equity criteria "in marginal populations".The article suggests that colonization in particular has led to the suppression of indigenous knowledge systems and ontology. The authors  note that colonialism has contributed to poverty, discrimination and racism which, 'are clearly important determinants of indigenous health and wellbeing, but are only intermediate causes........ Colonalization also constrains the design and diversity of potential climate and health responses, in particular through the suppression of indigenous knowledge systems and ways of being. Colonalization is itself just one manifestation of an exploitative Enlightenment philosophy that has resulted in catestrophic effects on indigenous peoples and nature."  (page 2 lines 68-73). The authors go on to present the example of the  Kaupapa Maori knowledge system as a theoretical approach derived from  indigenous people in New Zealand, that can be used in place of conventional western methodologies to assess the impact of climate mitigation on their health and well-being.  The authors indicate that the Kaupapa Maori knowledge system , " ...seeks to challenge the primacy of 'conventional' research methodologies  in order to centralize Maori ways of knowing and being".   Kaupapa  Maori  research methodology does not prescribe or prohibit and particular  study design or tools, but rather can utlize whatever methods are best suited to answer the research question(s)". (page 3 lines 98-101).

 Taking these aspects together, there appear to be several substantive weaknesses in the paper that the authors need to consider. These can be listed as follows.

  1. There is not enough detail (practical examples) of how colonialism (a system of government that has been exstinguished in practice for at least a quarter of a century in most countries around the world)  is purported to have influenced climate mitigation ( a relatively new concept and practice}.
  2. The Kaupapa Maori theoretical positioning is so amorphously defined as to make it difficult to understand what it is actually proposing in terms of methodology. What does it actually advocate methodologically if it does not prohibit any method which would presumably include conventional ones. And, how does one decide which methods are best suited to answer the research question(s)? As it stands it would seem that the Kaupapa Maori research methodology embraces all methodologies that are suited to answer specific reseach questions without putting forward any clear theoretical position.
  3. The article  moves from an inadequate definition of Kaupapa Maori research methodology, to an eclectic review of 39 studies which have focused on climate mitigation and inequity in marginal populations in countries throughout the world, without analyzing precisely how indigenous-centered research approaches would reduce co-harms of the colonalization effect-not even in the New Zealand situation where the Kaupapa Maori methodology might have been presented as a practical example.
  4. The study then concludes : "There is a dearth of information about the co-benefits and co-harms of climate mitigation policy for indigenous health and well-being. Much of the evidence that currentlly exists is from generic equity analysis and limited Western perspectives on relevant outcomes, which have serious limitations in relation to informing  future climate policy." ( page 13, lines 454-457). However, it is not clear at all how the authors have been able to draw this conclusion from the Kaupapa Maori theoretical positioning and analysis  presented in the study. This aspect needs further discussion and clarification. 

The study is nevertheless important in that is shows that many of the 'co-harms' of climate mitigation policies ride rough-shod over indigenous populations' ( and not necessarily marginal populations either, as the Africa  clearly shows) knowledge, social systems and use of natural resources traditionally needed for their on-going survival, health and well-being. The authors are  therefore encouraged to revise their paper accordingly.

Author Response

Response to Reviewer 4 comments

We would like to thank you very much for the helpful feedback on our manuscript. In the table below we have provided details of the revisions in the manuscript and explained our responses to the reviewer comments.

Reviewer comment

Authors’ response

1. There is not enough detail (practical examples) of how colonialism (a system of government that has been exstinguished in practice for at least a quarter of a century in most countries around the world)  is purported to have influenced climate mitigation ( a relatively new concept and practice}.

We dispute the suggestion that colonial forms of government have been extinguished in practice. We have identified the ongoing nature of colonization (lines 50-51) and described the ways in which colonization acts to differentially distribute health risks and benefits (lines 87-92).

In response to this comment, we have also strengthened one sentence in the article (lines 87-88) to be more explicit about the ongoing nature of colonization and its enduring impacts.

2. The Kaupapa Maori theoretical positioning is so amorphously defined as to make it difficult to understand what it is actually proposing in terms of methodology. What does it actually advocate methodologically if it does not prohibit any method which would presumably include conventional ones. And, how does one decide which methods are best suited to answer the research question(s)? As it stands it would seem that the Kaupapa Maori research methodology embraces all methodologies that are suited to answer specific reseach questions without putting forward any clear theoretical position.

Kaupapa Māori methodology is underpinned by Māori philosophies and Māori understandings of the world. It informs a set of principles that guide Kaupapa Māori research (e.g. see Curtis, E. Indigenous positioning in health research: the importance of Kaupapa Māori theory informed practice. AlterNative: An International Journal of Indigenous Peoples 2016, 12, 396.) Thus our Kaupapa Māori theoretical positioning does not prescribe or exclude any particular methods, but guides decisions about research methods and the ways in which they are applied.

We have added an explanation in the Methodology section (lines 141-144).

3. The article  moves from an inadequate definition of Kaupapa Maori research methodology, to an eclectic review of 39 studies which have focused on climate mitigation and inequity in marginal populations in countries throughout the world, without analyzing precisely how indigenous-centered research approaches would reduce co-harms of the colonalization effect-not even in the New Zealand situation where the Kaupapa Maori methodology might have been presented as a practical example.

We believe we have elucidated this in relation to the analysis of the three Indigenous-centred studies included in the review. One paragraph in the results section addresses these three studies and explains how their Indigenous methodologies contributed to distinctive insights and outcomes (lines 461-472).

We have added two sentences in the Discussion to further explain how Indigenous-centred approaches would contribute to decolonizing climate action (lines 574-578).

4. The study then concludes : "There is a dearth of information about the co-benefits and co-harms of climate mitigation policy for indigenous health and well-being. Much of the evidence that currentlly exists is from generic equity analysis and limited Western perspectives on relevant outcomes, which have serious limitations in relation to informing  future climate policy." ( page 13, lines 454-457). However, it is not clear at all how the authors have been able to draw this conclusion from the Kaupapa Maori theoretical positioning and analysis  presented in the study. This aspect needs further discussion and clarification.

Our Kaupapa Māori positioning meant that our validity assessment (lines 378-404) focused on how well studies addressed issues of importance to Indigenous peoples. This assessment highlighted limitations in regard to theoretical/methodological issues (e.g. consistency with Indigenous theoretical positioning), validity of the research process (e.g. involvement of Indigenous communities as investigators), and validity of outcomes (e.g. the extent to which potential impacts of particular relevance to Indigenous peoples were examined). For example, none of the studies examining food systems and dietary policies incorporated any consideration of customary Indigenous food sources or Indigenous food sovereignty practices. As a result of these findings, we concluded that there were serious limitations in relation to informing climate policy with respect to Indigenous peoples.

We have added a sentence in the Discussion highlighting this positioning and analysis as a strength of the research (lines 501-503).

The study is nevertheless important in that is shows that many of the 'co-harms' of climate mitigation policies ride rough-shod over indigenous populations' ( and not necessarily marginal populations either, as the Africa  clearly shows) knowledge, social systems and use of natural resources traditionally needed for their on-going survival, health and well-being. The authors are  therefore encouraged to revise their paper accordingly.

Thank you. We hope the revisions noted above have addressed this point.

Round 2

Reviewer 4 Report

You appear to have dealt with the concerns raised in my review acceptably. However, there are still concerns about the application of Kaupapa Maori methodology as a prescriptive methodology that advances the cause of 'culturally aware' climate mitigation policy.The review analysis indicates what aspects have been focused on and the aspects that involve ( quite loosely described) bits of Kaupapa Maori methodology without linking them up in  a tighter analytical framework indicating more precisely what aspect of the methodology ( if any) the various studies have considered and left out of consideration. A separate Table might be used for this purpose.

When the term 'colonization' occurs, it might be useful to distinguish the effects of colonization which have been enduring, from the physical process of colonization which effectively ceased in New Zealand some time ago.